# Integrating Stickers into Multimodal Dialogue Summarization: A Novel Dataset and Approach for Enhancing Social Media Interaction

Yuanchen Shi
Soochow University
School of Computer Science and Technology
Suzhou, China
20227927002@stu.suda.edu.cn

Fang Kong*
Soochow University
School of Computer Science and Technology
Suzhou, China
kongfang@suda.edu.cn

## Abstract

With the popularity of social media, growing number of online chats and comments are presented in the form of multimodal dialogues containing stickers. Automatically summarizing these dialogues can effectively reduce content overload and save reading time. However, existing datasets and works are either text dialogue summarization, or articles with real photos that respectively perform text summaries and key image extraction, and have not simultaneously considered the multimodal dialogue automatic summarization tasks with sticker images and online chat scenarios. To compensate for the lack of datasets and researches in this field, we propose a brand-new **M**ultimodal **C**hat **D**ialogue **S**ummarization **C**ontaining **S**tickers (MCDSCS) task and dataset. It consists of 5,527 Chinese multimodal chat dialogues and 14,356 different sticker images, with each dialogue interspersed with stickers in the text to reflect the real social media chat scenario. MCDSCS can also contribute to filling the gap in Chinese multimodal dialogue data. We use the most advanced GPT4 model and carefully design Chain-of-Thoughts (COT) supplemented with manual review to generate dialogues and extract summaries. We also propose a novel method that integrates the visual information of stickers with the text descriptions of emotions and intentions (TEI). Experiments show that our method can effectively improve the performance of various mainstream summary generation models, even better than some other multimodal models, ChatGPT, and Vision Large Language Models (VLMs). Our data and code are publicly available at https://github.com/FakerBoom/MCDSCS.

## CCS Concepts

• **Computing methodologies** → **Language resources**; *Discourse, dialogue and pragmatics*; Image representations.

* Corresponding author.

## Keywords

Multimodal Dialogue Summarization, Social Media Chat, Sticker Images, Visual Information

**ACM Reference Format:**
Yuanchen Shi and Fang Kong*. 2024. Integrating Stickers into Multimodal Dialogue Summarization: A Novel Dataset and Approach for Enhancing Social Media Interaction. In *Proceedings of the 32nd ACM International Conference on Multimedia (MM '24), October 28-November 1, 2024, Melbourne, VIC, Australia.* ACM, New York, NY, USA, 10 pages. https://doi.org/10.1145/3664647.3680978

## 1 Introduction

As social media platforms such as WeChat and TikTok continue to advance, people have become accustomed to inserting stickers into online conversations to express their thoughts [15]. Automated summaries of these multimodal dialogues can allow users to quickly obtain key information without reading through cumbersome chat logs or comment dialogues, thereby reducing memory usage.

However, due to the existence of various images, this task has become challenging. These images, commonly referred to as stickers, emoticons, emojis or memes[1], can vividly express the user's emotions [9] and behavioral intentions [28]. Therefore, it is necessary to consider both the dialogue text and stickers when summarizing. As shown in Figure 1, key contents reflecting speakers' feelings and actions such as "work out", "apology", and "go alone" can only be gleaned from the stickers. Without the stickers, it is impossible to create a complete summary without missing crucial information.

Yet, existing multimodal summarization tasks are mostly confined to news [19], teaching [27] or open domain [22] video summaries, or separate summaries of articles and images [44, 45]. There is little research on generating text summaries from both text and images. Not to mention in dialogue scenarios, which makes multimodal dialogue summarization even more difficult due to the specificity of data format, colloquialism and arbitrariness of text and images, and scarcity of suitable datasets.

To fill the gap in this field, we propose the Multimodal Chat Dialogue Summarization Containing Stickers (MCDSCS) task, which focuses on researching automatic summarization of chat dialogues on social media that include text and stickers. We also propose MCDSCS dataset for research on this task. First, we select informative and meaningful sentences from a large amount of social media chat text, use suitable COTs [38] to generate dialogues with GPT4 from these sentences, and carry out automatic and manual

---

[1]In this paper, we collectively refer to them as "stickers".

screening. Afterwards, we download over 30k sticker images from various social apps, as well as from websites like Baidu and Microsoft, then manually select and insert them into dialogues by adding or replacing text. Finally, we use the most advanced GPT4-Vision model, combined with automatic and manual reviews, to get summaries that take into account multimodal information. The resulting MCDSCS dataset, as shown in Figure 1, consists of 5.5k multimodal dialogues containing Chinese chat text and stickers. With the exception of a few short dialogues, most dialogues contain two or more stickers, with a total of 14.4k different sticker images.

Meanwhile, we propose a new method, which adds **T**ext descriptions of **E**motions and **I**ntentions (TEI) of stickers, to handle multimodal dialogue summaries. It integrates sticker information by inserting text descriptions that show the emotions and intentions expressed by the stickers into the original dialogues. Compared to methods that only consider dialogue text or multimodal models which integrate visual features, our method enables baseline models such as the zero-shot LLM (ChatGPT) and fine-tuned models (Bart, T5, etc.) to achieve better performance, which is even better than that of Vision Large Language Models (VLMs).

Our main contributions are as follows:

- We propose a new task MCDSCS. To the best of our knowledge, it is the first to perform text summarization on multimodal dialogues including stickers.
- We propose the MCDSCS dataset, the first multimodal Chinese dialogue summary dataset that includes a large number of sticker images.
- We propose a novel method, TEI, which outperforms other methods, models and VLMs on the MCDSCS task.

## 2 Related Work

### 2.1 Dialogue Summarization

At present, most dialogue summarization dataset is text-based. Gliwa et al. [11] constructed a dataset SAMSum by manually selecting and annotating available chat conversations. Chen et al. [4] extracted over 13k dialogues from online English listening and speaking exams and manually annotated the summaries. MEDIASUM [43] collected extensive media interview transcripts from NPR and CNN, the summaries and topic descriptions of which were used as summaries for each dialogue. In terms of Chinese dialogue summarization, CSDS [21] excerpted some dialogue from JD customer service dataset JDDC [3], and summaries were made from both the user and customer service perspectives. They all collected existing dialogues in text form and obtained summaries manually.

Currently there are many effective methods for text dialogue summarization. Most popular generative models have been researched for dialogue form text summarization and have achieved good results [6, 17, 24, 37]. Khalifa et al. [16] explored four different challenges of dialogue summarization: handling and differentiating parts of the dialogue belonging to multiple speakers, negation understanding, reasoning about the situation, and informal language understanding. Liu et al. [23] explicitly incorporated co-reference information into summarization models, solving the difficulties brought about by complex co-reference links in the dialogue. However, these methods can't effectively solve the task of dialogue summarization that involve images.

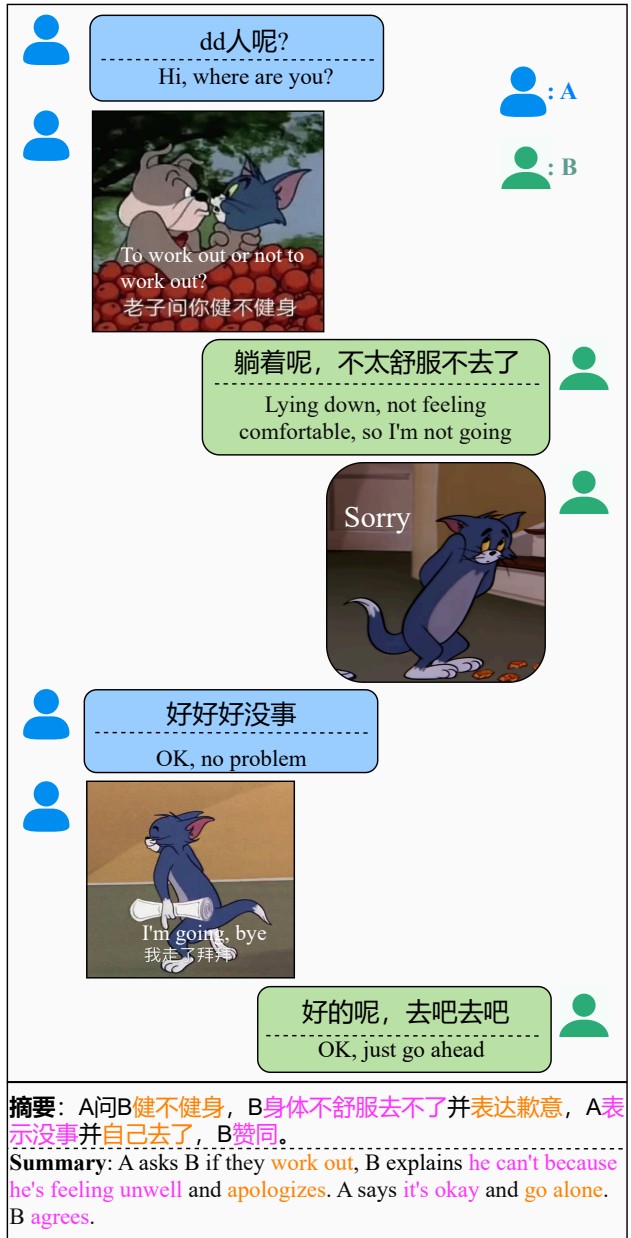

**Figure 1: An example from the MCDSCS dataset. The English below the dashed line in the text and in the image are corresponding translations of the Chinese. The pink and orange text in the summary corresponds to contents that can only be obtained from the text and stickers, respectively.**

### 2.2 Multimodal Summarization

The multimodal field has become increasingly popular in recent years. Video summarization tasks in various fields have received extensive research [1, 19, 22]. The summarization tasks of text and images are mostly in the form of multimodal summarization with

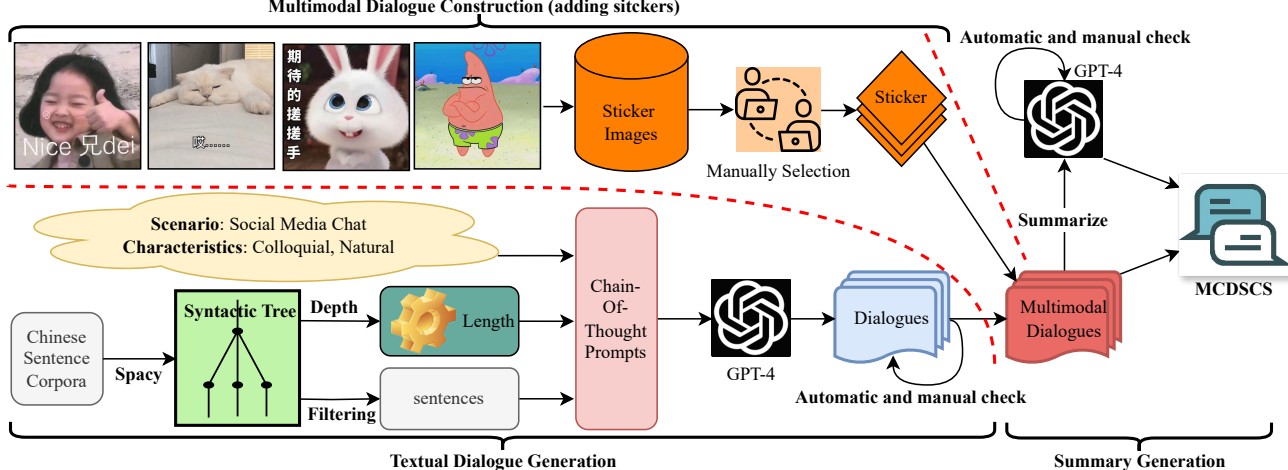

**Figure 2: The overall process of constructing the MCDSCS dataset, including stages such as textual dialogue generation, multimodal dialogue construction, and summary generation, along with automatic and manual checks, which are delineated by red dashed lines.**

multimodal output (MSMO) [44], obtaining the summary of the text and the most important image among all. Zhu [45] proposed a multi-modal target function guided by multi-modal references, utilizing the loss of summary generation and image selection, and proposed a novel evaluation metric based on a joint multi-modal representation for MSMO. HCSCL [41] is a Hierarchical Cross-Modality Semantic Correlation Learning Model for learning the intra-modal and inter-modal correlations existing in multimodal data, along with a new dataset for the MSMO task, which includes related image annotations and image object label information.

Different from MSMO, the MCDSCS task requires a text summary as output to summarize the multimodal content, which is more concise and suitable for online conversation scenarios. The How2 dataset [32] is a multimodal video tutorial collection with text subtitles and translations. Palaskar et al. [27] used the text and video frames in the How2 dataset to generate text summaries, which is similar to our task. The MREDDITSUM dataset [26] includes posts with images and text comments on the Reddit website, and proposes a cluster-based multi-stage summarization (CWS) method to generate text summaries of post content and images. However, since the images, content, and comment text in the posts can be independent of each other, and the text and images in dialogues are continuous and interrelated, the CWS method of staged summarization is not applicable to the MCDSCS task.

Furthermore, the sticker images in MCDSCS come in a variety of styles and contain a lot of metaphors and irony, which significantly differ from the real photos and videos in other multimodal datasets. This also brings considerable challenges.

## 3 The MCDSCS Dataset

### 3.1 Dataset Construction

As shown in Figure 2, we outline the complete process of constructing the MCDSCS dataset, starting from sentences to textual

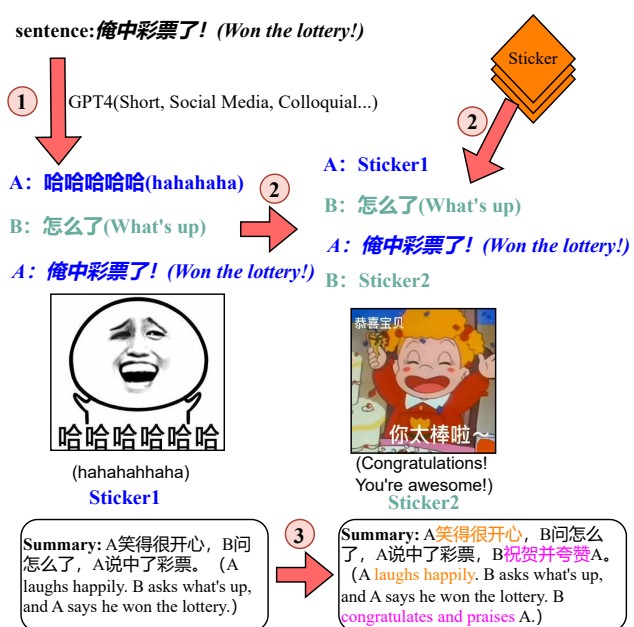

**Figure 3: An example of dialogue generation (step 1) and sticker insertion (step 2), and how stickers can impact the summary (step 3). The English translations are provided in brackets. In the summary, orange text corresponds to information that can be obtained from both the original text and the replaced sticker, while pink text corresponds to content that can only be derived from the newly added sticker.**

dialogues, further to multimodal dialogues, and ultimately obtaining the overall summary.

**Table 1: Comparison of MCDSCS with other summarization datasets. # stands for average, *Len.* stands for Token length, *T.C.Rate* stands for text compression rate, and within *Modality, t* denotes text, *i* denotes image, and *v* represents video. *sens.* stands for sentences. The How2 dataset doesn't calculate a compression rate because it only has the number of subtitle sentences to refer to. *Auto.Gen* indicates whether the text and summary in the dataset are auto-generated.**

| Dataset | Domain | Language | Size | #Turns | #Text Len. | #Sum Len. | T.C.Rate(%) | Modality | Auto.Gen |
|---|---|---|---|---|---|---|---|---|---|
| MCDSCS(ours) | Dialog | ZH | 5527 | 12.69 | 71.73 | 25.27 | 35.23 | t,i | Yes |
| SAMSum | Dialog | EN | 16396 | 9.78 | 124.10 | 23.40 | 18.86 | t | No |
| CSDS | Dialog | ZH | 10701 | 25.92 | 321.93 | 70.83 | 22.00 | t | No |
| MREDDITSUM | Forum | EN | 3033 | 22.6 | 691.00 | 91.00 | 13.17 | t,i | No |
| How2 | Video | EN | 79114 | 1 | 14.00(sens.) | 33.00 | - | t,v | No |
| MSMO DailyMail | News | EN | 3144581 | 1 | 720.87 | 55.00 | 7.63 | t,i | No |

*3.1.1 Sentence-To-Dialogue.* We collect over 15k Chinese chat sentences from the CSMSA [10] and xiaohuangji [2] datasets, both of which contain a large number of Chinese chat sentences. In order to obtain rich and meaningful dialogues, we require that the source sentences should not be overly simple and should contain sufficient information. We construct a syntactic dependency tree [36] for each sentence to select sentences that are rich enough in content to form a dialogue. Any sentence with a syntactic tree depth less than 2 is considered to lack sufficient content for dialogue construction and filtered out. In this way, we obtain about 7k sentences.

We decide the number and length of dialogue turns based on the depth of the syntactic tree of sentences, such as 3 corresponds to short and 5 corresponds to medium. See Section 1 in Supplementary for detailed statistics. Then we add "Social Media" to the COT to control dialogue scenario, and incorporate characteristics of chat such as "Colloquial" and "Natural". We call on the API of GPT4 to generate dialogue for each sentence. We also add detection of whether it contains sensitive or violative information, and automatically filter out inappropriate dialogues during the generation process. For the generated dialogues, we ask GPT4 whether the dialogue has logical errors and whether the content is in line with social media chat. We manually modify or filter out dialogues that are illogical or poorly evaluated, so as to minimize the hallucination issues that LLM may cause. Eventually, after multiple rounds of screening, we obtain 5.5k chat dialogues. Figure Section 2 in Supplementary shows a typical COT example.

*3.1.2 Dialogue-To-Multimodal-Dialogue.* We download a total of more than 30k stickers from software like QQ, WeChat, TikTok, as well as from Baidu and Microsoft search engines. We employ five Chinese language professionals who have experience with dataset annotation. Each annotator is randomly assigned 1.1k dialogues and 6k stickers. They are instructed to add at least two stickers to each dialogue with five rounds or more, and to try to avoid duplicate stickers to increase the diversity of stickers. The method of inserting stickers is shown in step 2 in Figure 3. Sticker1 represents replacing the original text with a corresponding sticker, and Sticker2 indicates adding a suitable sticker directly while maintaining the coherence and fluency of the dialogue. After inserting the stickers, the five annotators exchange the multimodal dialogues they annotated and communicate to modify any inappropriate stickers. Finally, within a

---
[2]https://github.com/aceimnorstuvwxz/dgk_lost_conv/tree/master/results

**Table 2: Sticker quantity (*Qua.*) and percentage (*P*) statistics. *nS* represents the number of dialogues containing n stickers. *First, Last, Mid* respectively represent that stickers appear in the first sentence, last sentence, and middle of the dialogue.**

| | 1S | 2S | 3S | 4S | 5+S. | First | Mid | Last |
|---|---|---|---|---|---|---|---|---|
| Qua. | 465 | **2573** | 1455 | 662 | 372 | 155 | **5480** | 947 |
| P(%) | 8.41 | **46.55** | 26.33 | 11.98 | 6.73 | 2.80 | **99.15** | 17.14 |

total of 5.5k dialogues, we add stickers 14.7k times. After excluding duplicate stickers from the statistical data, a total of 14.4k distinct stickers are used.

*3.1.3 Summary Annotation.* We call the GPT4-Vision-Preview's API to generate a summary that encapsulates the dialogue containing text and stickers. We then ask GPT4 whether it have taken into account the information from stickers, and give the obtained feedback and the original data back to the annotators simultaneously to determine if the summary needs adjustment. The results show that less than 3% of the data (158 pieces) need manual summary rewriting, mostly because the input stickers don't comply with openai policy. For this small portion of data, our annotators manually write multimodal summaries referring to the rest auto-generated summaries, and make modifications based on suggestions from the other four annotators. It also verifies that our obtained summaries nicely blended the key information from each modality. Our COT for getting summaries and a detailed example are shown in Section 3 in Supplementary.

*3.1.4 Typical Example.* Figure 3 shows a typical example of how we construct the data. In step 1, we generate a dialogue from a single sentence using GPT4 and incorporate constraints on dialogue length, scenario, and characteristics through COT. Step 2 shows how we add stickers to construct multimodal dialogues. Step 3 illustrates the impact of adding stickers on the summary. "laugh happily" can be derived from both the sticker and the original text, while "congratulate and praise" is additional multimodal information introduced after inserting stickers, which differs from the summary of the original textual dialogue. This highlights the importance of stickers for the multimodal summarization and underscores the challenge and value of the MCDSCS task.

**Table 3: The comparison of Extractive-Oracle ROUGE Scores. The lower the score, the more abstractive the summaries are. *EOR-n* stands for Extractive-Oracle ROUGE-n.**

| Dataset | EOR-1 | EOR-2 | EOR-3 |
|---|---|---|---|
| MCDSCS(ours) | 17.91 | 6.52 | 16.89 |
| MREDDITSUM | 36.52 | 11.95 | 31.42 |
| AnswerSumm | 40.05 | 18.45 | 35.70 |
| ConvoSummreddit | 35.74 | 10.45 | 30.74 |

## 3.2 Data Statistics

*3.2.1 Overall Statistics.* The data statistics of MCDSCS is shown in Table 1. We divide 5527 pieces of data into 4421 for the train-set, and 553 each for the dev-set and test-set. Compared with other datasets, we are the only multimodal dialogue summarization dataset, especially in Chinese, which are even more scarce than English datasets. Due to privacy protection and the difficulty in preservation, it is hard to excerpt chat dialogues from social media on a large scale. Therefore, the scale of MCDSCS is not large compared to other dialogue and news datasets. However, it is the only dataset that utilizes auto-generation for dialogue and summary acquisition. This method is very flexible and can be applicable to the construction of many kinds of datasets, saving a lot of time on manual collection and summarization. The average text length of MCDSCS is the shortest because many contents expressed in text are replaced by stickers, and it also aligns with the concise and colloquial characteristics of social media chats. This also explains why the text compression rate of MCDSCS is relatively high, as the summary still contains a lot of information derived from analyzing stickers.

*3.2.2 Sticker Statistics.* Table 2 illustrates the distribution of stickers in our dataset. Over 90% of dialogues contain two or more stickers, with the maximum number of stickers in a dialogue being 11. We also research the occurrence positions of stickers in dialogues. It can be observed that a minority of dialogues start with a sticker, a certain number of dialogues end with a sticker, while almost all dialogues feature stickers interspersed throughout the conversation. The above statistics also reflect to some extent the real usage of stickers in social media platforms.

## 3.3 Quality Analysis

*3.3.1 Abstractiveness Assessment.* Extractive-Oracle ROUGE, as a metric for measuring the Abstractiveness of summaries, has been used in many summarization datasets. We list the comparison of Extractive-Oracle ROUGE values between the MCDSCS dataset and several datasets in Table 3. The scores for these datasets all come from their respective papers [7, 8, 26]. Our dataset is far more abstractive than others, which is a significant feature of auto-generated summaries, and also demonstrates that much of the content in our summaries comes from sticker images. This also aligns well with the casual nature of social media chats.

*3.3.2 Image Correlation.* We randomly select 300 images from stickers used by each annotator, totaling 1500 stickers. We calculate the CLIPScore and RefCLIPScore [13] for each image with their

**Table 4: The CLIPScore and RefCLIPScore scores between the sticker images and the context text and summary (*summ.*). *Con.-n* represents n sentences of the context above and below the sticker. *Ran.* represents the average score of randomly sampled alternative sticker replacements for the current one.**

| Metrics | Con.-1 | Con.-2 | Con.-3 | Summ. |
|---|---|---|---|---|
| CLIPScore | 62.33 | 62.43 | 62.33 | 60.47 |
| RefCLIPScore | 74.39 | 74.36 | 74.33 | 72.77 |
| Ran.-CLIPScore | 62.14 | 62.23 | 62.19 | 60.34 |
| Ran.-RefCLIPScore | 74.23 | 74.22 | 74.23 | 72.67 |

**Table 5: Human evaluation of datas in different summarization datasets. *Flu., Con., Rel., Coh.* stand for Fluency, Consistency, Relevance, and Coherence, respectively.**

| Dataset | Content | Flu. | Con. | Rel. | Coh. |
|---|---|---|---|---|---|
| MCDSCS | dialogue | **4.96** | 4.61 | 4.73 | **4.83** |
| | summary | **4.85** | **4.88** | **4.98** | **4.85** |
| CSDS | dialogue | 4.78 | **4.66** | **4.89** | 4.25 |
| | summary | 4.20 | 4.80 | 4.97 | 4.23 |
| MREDDITSUM | post | 4.21 | 3.86 | 4.59 | 4.44 |
| | summary | 4.13 | 4.32 | 4.82 | 4.54 |

accompanying 1, 2, and 3 sentences of context, as well as the generated summary. CLIPScore, leveraging the CLIP model [29], compute the relevance between text and images. RefCLIPScore, on top of this, also calculate similarity, adaptability between images and text, and their relevance with the provided references. We utilize the TEIs of stickers obtained in Section 4.3 as reference text for RefCLIPScore. We also randomly replace each sticker with 100 other stickers and calculate the scores with the current context and summary.

Table 4 shows the scores, which indicates that the stickers we add have a close and logically interwoven relationship with their accompanying contextual text. Although the scores with the summary is slightly lower, considering that the summary contains more than one sticker and textual information, such results are still good. The high RefCLIPScore further confirms the close connection between the TEIs of images and the contextual dialogue and summary. When stickers are replaced by other random stickers, all scores decrease. This also indicates the excellent compatibility between the stickers added to our dataset and the original dialogues.

*3.3.3 Human Evaluation.* Similar to prior works, we manually evaluate the Fluency, Consistency, Relevance, and Coherence of MCDSCS, the Chinese dialogue summarization dataset CSDS, and multimodal post dataset MREDDITSUM, of which the task format is most similar to ours. We divide the evaluation into dialogues (posts) and summaries within the dataset. Fluency assesses the smoothness of each individual sentence. Consistency evaluates whether the dialogue (post) is consistent or contradictory and whether the summary aligns with the original data. Relevance assesses the relevance

of the dialogue (post) to its scenario and whether the summary content is pertinent to the original data. Coherence evaluates the overall quality and logicality of all text and images.

Considering the different sizes of the datasets, we randomly select 200 pieces of data from each dataset. We employ 30 graduate students majoring in literature to individually score these data on the four metrics, with scores ranging from 1 to 5. A higher score indicates better quality, up to a maximum of 5 points. Each reviewer is a Chinese speaker and has a good level of English and has passed the CET-6 exam with high scores.

Table 5 shows the average human scores. Our dialogues and summaries exhibit the highest fluency and coherence, which reflects that the text in MCDSCS is fluent and the images are appropriate, effectively reducing the awkward chats and irrelevant responses in dialogues. Since CSDS consists of e-commerce customer service dialogues, the content tends to be singular, resulting in higher scores for consistency and relevance. However, the consistency and relevance of our summaries of multimodal dialogues are also excellent. This is also due to the powerful multimodal performance of GPT4, which makes the quality of the summaries very high. MREDDITSUM scores lower on fluency and consistency because posts and comments on the site often feature internet slang, and images, posts, and comments are relatively independent of each other, frequently resulting in disordered comment replies.

## 4 Experiments

### 4.1 Task Description

The MCDSCS task, as shown in Figure 1, requires a multimodal multi-turn dialogue as input, where each turn may consist of either a text sentence or a sticker image. The model is then required to output a single text sentence summarizing the key content of all the text and stickers in the entire dialogue.

### 4.2 Baseline Models

*4.2.1 Extractive Algorithms.* We use the following commonly used summary extraction algorithms to extract summaries:

**LEAD-***n* [33] selects the first *n* utterances. **LONGEST-***n* [11] selects the longest *n* utterances. **EXT-ORACLE-***n* [25] can be considered as an upper bound on algorithmically extracted summaries by selecting the *n* utterances with the highest ROUGE scores computed against the summary. **LexPageRank** [5] ranks dialogue utterances by PageRank algorithm and extracts utterances in order until the length of the summary reaches the limit.

In LEAD, LONGEST, and EXT-ORACLE, we set *n* as 3, 3, and 2, respectively, which are chosen after experimenting with *n* ranging from 1 to 5 to find the optimal selections.

*4.2.2 Text-only Generative Models.* We select several pre-trained language models widely used for text summarization: **BART** [18] is an encoder-decoder Transformer model pretrained on a large corpus using a denoising autoencoder task. We use the large version of BART. **T5** [30] is a versatile pre-trained transformer-based model, formulating language tasks as a unified text-to-text problem. We use the base version of T5. **Pegasus** [40] is a pre-trained abstractive text summarization model, employing a pre-training objective called "Gap Sentences". We use the large version of Pegasus. **LLaMA** [35] is

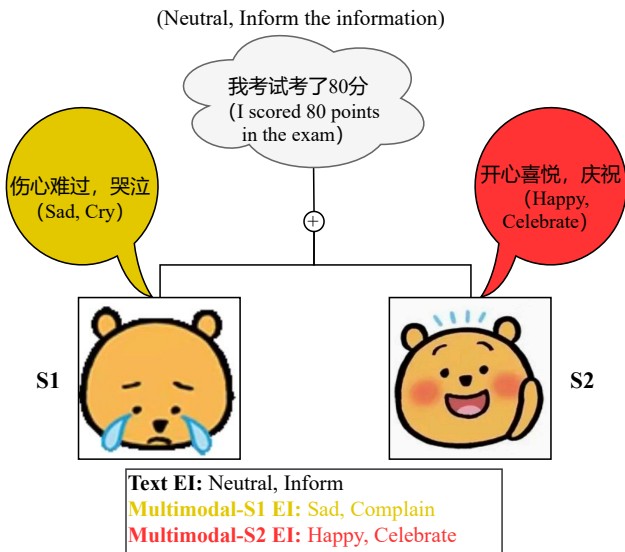

**Figure 4: An example to illustrate how stickers can influence the multimodal emotion and intention (*EI*). The English translations are provided in brackets.**

a collection of foundation language models ranging from 7B to 65B parameters. We use the LLaMA-v2-7B version with the Low-Rank Adaption (LoRA) [14].

*4.2.3 Vision-Guided Generative Models.* Vision-Guided Bart and T5 (VG-Bart and VG-T5) [39] are used for the How2 video multimodal summarization task, with an extra visual layer added to the model to handle video embeddings. We use the CLIP model to extract feature embeddings for all stickers as a substitute, and train the visual layer using the sticker images and their text descriptions. Experimental setups for finetuned models are in Supplementary.

*4.2.4 Zero-Shot LLM.* We select ChatGPT (GPT3.5), for zero-shot experiments, since ChatGPT has achieved remarkable results in most natural language processing (NLP) tasks and benchmarks [34].

*4.2.5 Zero-Shot VLMs.* We select LLaVA [3], Yi [4] and Qwen [5] as the VLM baselines. The model versions and experimental methods are described in Section 4 in Supplementary. Since VLMs can independently process sticker images, our TEI method is not applicable, so they are only used for comparison experiments.

### 4.3 TEI Method

Sticker images can succinctly and vividly convey the speaker's emotions and intentions [10, 31]. The same text, paired with different stickers, can convey completely different overall information, as illustrated in Figure 4. Simply looking at the text doesn't reveal the speaker's emotional inclination. However, when coupled with an sticker representing sadness, the multimodal content can be analyzed to see that it's complaining getting only 80 points in a

---

[3] https://github.com/haotian-liu/LLaVA
[4] https://github.com/01-ai/Yi
[5] https://github.com/QwenLM/Qwen

**Table 6: Overall experimental results. *TEI* represents our method, which is adding the text descriptions of emotions and intentions of stickers to dialogues.**

| Model | R1 | R2 | RL | Bl. | Met. |
|---|---|---|---|---|---|
| *Extractive* | | | | | |
| LEAD-3 | 24.15 | 8.53 | 20.11 | 7.41 | 9.75 |
| LEAD-3-TEI | 23.52 | 7.51 | 18.54 | 7.86 | 11.04 |
| LONGEST-3 | 28.13 | 10.20 | 21.08 | 8.69 | 12.84 |
| LONGEST-3-TEI | 19.14 | 4.36 | 12.90 | 8.01 | 11.90 |
| EXT-ORACLE-2 | 33.92 | 14.67 | **25.61** | 8.12 | 13.84 |
| EXT-ORACLE-2-TEI | 34.32 | 14.25 | 24.89 | 9.15 | 15.81 |
| LexPageRank | 34.47 | 14.69 | 25.12 | 10.40 | 15.63 |
| LexPageRank-TEI | **35.53** | **15.43** | 25.52 | **10.79** | **16.95** |
| *Fine-tuned Generative* | | | | | |
| Bart | 44.07 | 21.17 | **38.11** | 14.60 | 29.74 |
| Bart-TEI(ours) | **45.67** | **22.38** | 36.86 | **15.09** | **30.22** |
| VG-Bart | 45.58 | 21.90 | 35.23 | 14.09 | 28.44 |
| T5 | 31.82 | 13.49 | 26.91 | 9.14 | 16.24 |
| T5-TEI(ours) | 31.99 | **13.89** | **27.22** | **9.73** | **16.86** |
| VG-T5 | **32.05** | 13.20 | 26.89 | 8.95 | 16.14 |
| Pegasus | 43.08 | 18.35 | 36.08 | 11.79 | 26.87 |
| Pegasus-TEI(ours) | **44.24** | **19.80** | **37.26** | **12.58** | **28.31** |
| LLaMA2 | 52.31 | 25.00 | 42.42 | 13.69 | 39.21 |
| LLaMA2-TEI(ours) | **55.74** | **36.07** | **46.87** | **15.55** | **44.45** |
| *Zero-Shot LLM* | | | | | |
| ChatGPT | 38.26 | 15.32 | 31.69 | 9.48 | 20.97 |
| ChatGPT-TEI(ours) | **41.57** | **18.85** | **35.42** | **10.03** | **21.88** |

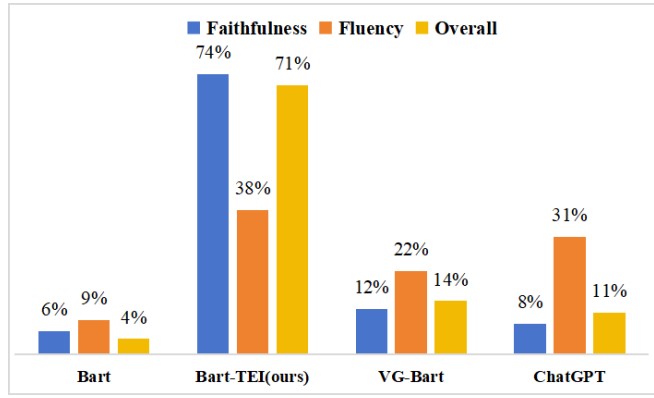

**Figure 5: Human evaluation results of randomly sampled summaries of different models.**

melancholic way. Similarly, when paired with a happy and joyful sticker, it can be concluded that the statement is celebrating the high score of 80 with happiness. This inspires us to decipher the emotions and intentions expressed by stickers.

We input stickers into the GPT4-Vision model, asking it to describe the sticker in terms of emotions and intentions it expresses. We replace the stickers with these descriptions in dialogues.

We calculate the CLIPScore of all stickers and their TEI. Considering the conciseness of TEI, which typically consists of only a few words, the result (62.12) is already quite commendable, proving that the TEIs we got are closely related to the stickers.

### 4.4 Evaluation Metrics

We refer to previous summarization work and use ROUGE scores (ROUGE-1, ROUGE-2, ROUGE-L) [20], Bleu-score with smoothing_function [12] and Meteor-score [2] [6] as evaluation metrics.

## 5 Results and Analysis

### 5.1 Extractive Results Analysis

From Table 6, it can be seen that the TEI method does not perform well for the LEAD and LONGEST algorithms, since these two algorithms use formalized methods to extract sentences and do not consider the text content. While the TEI is actually aimed at enhancing the understanding of the image and related to the

---
[6]Next, we will use R1, R2, RL, Bl and Met as abbreviations.

final summary, direct copying is not advantageous. For the EXT-ORACLE and LexPageRank, these two algorithms approximate the upper limit of extractive summary. Because these methods compute the relationship between the text and the final answer, their extraction is relatively intelligent and can filter out some irrelevant text. Thus, the results are improved. Extractive results are generally superior, reflecting that the summaries in our dataset closely align with the original dialogue text and stickers.

### 5.2 Generative Results Analysis

Table 6 shows the scores of all generative models including Chat-GPT. It can be seen that adding TEI to each model will improve its performance significantly, making it superior to the baselines. Moreover, after incorporating TEI, performance of the Bart and T5 models outperform their own Vision-Guided models that has added visual layers and image features. We hypothesize that this is because directly embedding image features may introduce noise brought about by the background, lines, and irrelevant texts. In contrast, after being converted into TEI, the model can more easily understand the multimodal emotions and intentions from a textual perspective. In addition, apart from the base version of the T5 model, which may have a smaller model size and fewer parameters, the performance of other models after fine-tuning with TEI greatly surpasses that of ChatGPT, even though the performance of ChatGPT also significantly improves with TEI. This indicates that the TEI method is also highly effective in zero-shot scenarios. Additionally, comparing the experimental results of VLMs in Supplementary shows that VLMs do not perform well in multimodal dialogue summarization tasks involving stickers. This further demonstrates the challenging nature of the MCDSCS dataset.

The overall results show that replacing stickers with TEI in dialogues yields effectiveness, indicating that the main role of stickers in social media is to express user's emotions and intentions. TEI is deemed to be an appropriate approach to fuse stickers and text.

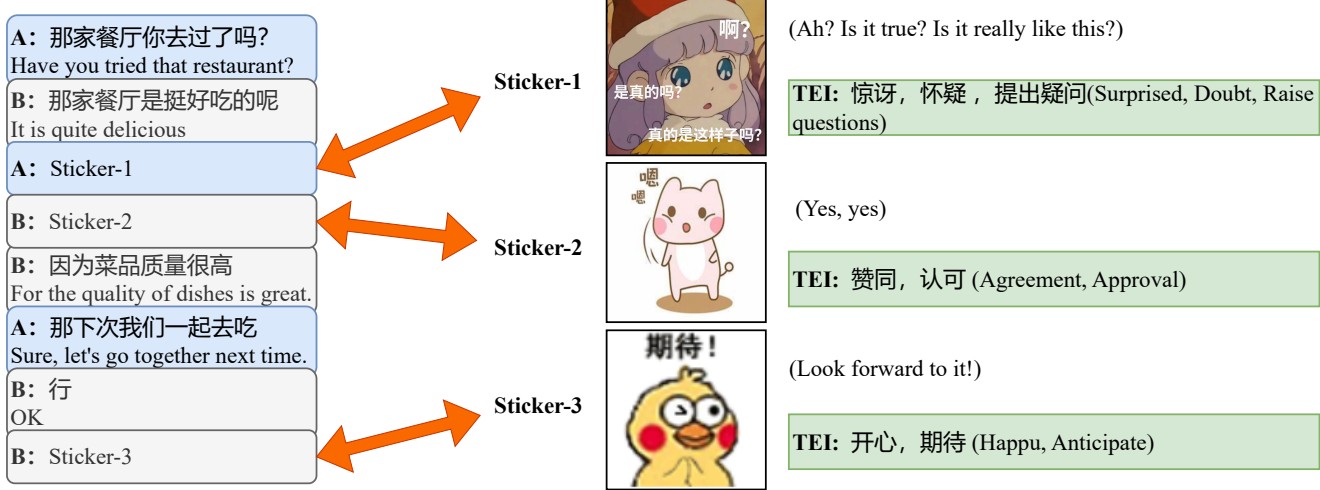

**Figure 6: A typical example of summaries obtained from our fine-tuned Bart series models and ChatGPT. The purple font represents visual information that aligns with the TEI of stickers. The English translations are below Chinese sentences.**

## 5.3 Case Study

Figure 6 displays summaries generated by some well-peoformed models for one instance in MCDSCS. The Bart model only summarizes the text, omitting all visual information and neglecting the emotional tendencies of speakers. The result significantly improves after incorporating TEI, which takes into account the visual information. VG-Bart mainly introduces visual features. Compared to the questioning expressed through text in Sticker-1, VG-Bart places more emphasis on the character's surprised expression. Also, it overlooks the intention conveyed by Sticker-2 and Sticker-3. Chat-GPT can fluently obtain a complete summary of the dialogue text, but it also misses the multimodal information of stickers.

We randomly select 30 instances and corresponding summaries generated by the different models. We recruit 100 undergraduate students majoring in Chinese Language and Literature to vote to select the summary they believe that best met the evaluation criteria for each instance. Referring to existing work [26, 42], we choose fluency, faithfulness, and overall as the evaluation criteria. Fluency represents whether the summary is fluent and natural, faithfulness measures the extent to which the summary conforms

to the dialogue, and overall represents the overall user preference. As shown in Figure 5, the summaries generated by our method received the majority of preferences in terms of faithfulness and overall quality. At the same time, it also outperforms other methods in terms of fluency. We believe this indicates that our method can get summaries that are more consistent with social media scenarios and better reflect the true feelings of speakers.

## 6 Conclusion and Future Work

The use of stickers in social media dialogues is increasing. To address this, we propose the MCDSCS task and dataset, comprising 5.5k multimodal dialogues, 14.4k sticker images, and reviewed auto-generated text summaries. We also introduce the TEI method, which integrates text descriptions of stickers' emotions and intentions into dialogues, outperforming baseline models, multimodal models, and ChatGPT. Future work includes applying our dataset construction method to create multimodal dialogue datasets in various languages and tasks, and developing improved multimodal fusion methods for better sticker information extraction and integration.

## Acknowledgments

This work was supported by the Project 62276178 under the National Natural Science Foundation of China, the Key Project 23KJA520012 under the Natural Science Foundation of Jiangsu Higher Education Institutions and the Priority Academic Program Development of Jiangsu Higher Education Institutions.

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
