# OpenReview forum: "Integrating Stickers into Multimodal Dialogue Summarization: A Novel Dataset and Approach for Enhancing Social Media Interaction"
_acmmm.org/ACMMM/2024/Conference — MM2024 Poster_

### Official Review · Reviewer_9wao · 2024-05-15

**Rating:** 4
**Confidence:** 4

**Summary:**

This paper proposes a brand-new Multimodal Chat Dialogue Summarization Containing Stickers (MCDSCS) task and dataset. At the same time the authors also propose a novel method that integrates the visual information of stickers with the text descriptions of emotions and intentions (TEI). Experiments show that the proposed method can improve the performance of various mainstream summary generation models.

**Strengths:**

(1）It takes a lot of work to construct a new multimodal Chinese dialogue summary dataset MCDSCS.
(2) This paper proposes a baseline model to evaluate the dataset MCDSCS.

**Limitations:**

(1) The research is mainly based on Chinese social media platforms, and the stickers may contain specific cultural backgrounds and contexts, which may limit the generalization ability of the model and its application in other languages or cultural contexts.
(2) The paper proposes a new method for integrating visual information, but does not compare in detail with other existing methods of multimodal conversation summarization, especially those that also consider image and text information. The specific comparison with other mainstream multimodal dialogue summarization technologies, especially those that integrate image processing, should be described in detail, and the advantages and innovations of this method should be demonstrated.
(3) The article is too general in describing the technical details of the algorithm and model, and lacks the necessary theoretical support and explanation of parameter settings.

**Suitability:**

3

---

### Official Review · Reviewer_tqqY · 2024-05-24

**Rating:** 4
**Confidence:** 3

**Summary:**

This is a paper focusing on the multimodal chat dialogue summarization task that integrates sticker information. The paper proposes a new task of multimodal chat dialogue summarization containing stickers  and contributes a corresponding dataset. In the dialogue, the intent and emotional information contained in the stickers can help understand the overall intent of the chatters and the emotional direction of the conversation. This part of the information is a very important part of the input for chat dialogue summarization. The perspective of the paper is interesting and reasonable.

**Strengths:**

1.The paper's approach of enhancing multimodal dialogue summaries with the intent and emotional information of multimodal dialogue stickers is novel.
2. The dataset provided by the paper contributes to the research of tasks related to this field, and the statistical introduction and analysis of the dataset are clear and comprehensive.
3.The framework and dataset of this task can be extended to various social multimodal interaction scenarios involving stickers, such as posting and replying on social platforms with sticker images.

**Limitations:**

1.The dataset analysis is thorough, and the motivation for using stickers' intent and emotional information is sound. However, the methodology is comparatively straightforward and lacks distinctive features.  The authors could summarize the essence and innovation of the method in 1-2 sentences; currently, it appears to simply describe stickers information obtained from GPT-4.
2. The models used in sections 4.2.2 and 4.2.3 seem relatively outdated. The performance of some recent open-source llms on this task should be more noteworthy compared to models like BART and T5.

**Suitability:**

3

---

### Official Review · Reviewer_PyhX · 2024-05-28

**Rating:** 3
**Confidence:** 3

**Summary:**

This paper proposed a novel task and dataset, namely Multimodal Chat Dialogue Summarization Containing Stickers (MCDSCS). The dataset is generated using the GPT-4 model through chain-of-thoughts design and manual review to produce dialogues and extract summaries. Additionally, this paper proposes a novel approach that combines the visual information of stickers with text descriptions of emotions and intentions (TEI) to guide the language model in better adapting to the multimodal summarization task. Experimental results show that TEI improves the performance of the summary generative model in multiple metrics.

**Strengths:**

1) The paper is simple and easy to understand.
2) The proposed method improves performance across multiple metrics.

**Limitations:**

Weakness:
1) The Chinese summary and the English summary in Figure 1 do not convey the same meaning.
2) From Figure 1 and Table 1, it is evident that the quality of the proposed dataset is not high.
3) There is insufficient analysis corresponding to the data in Table 2, and the information that can be derived from Table 2 is unclear.
4) The abstractive in Section 3.3 refers to what, and why is it considered that the more abstract the text, the better? Section 3.3 does not explain this.
5) In Table 5, why are Con. and Rel. bolded in the CSDS dataset?
6) In Section 4.3, what does TEI stand for? Why is there a mention of "their TEI" in line 692?
7) What does "the result (62.12)" mean in line 694?

Typos:
1. Line 495: Abstractiveness -> abstractiveness.
2. Figure 2: sentences->Sentences, Chain-Of-Thought Prompts ->Chain of Thought Prompts.

**Suitability:**

1

---

### Official Review · Reviewer_Vqn3 · 2024-05-28

**Rating:** 4
**Confidence:** 2

**Summary:**

This paper proposes a new task called Multimodal Chat Dialogue Summarization Containing Stickers (MCDSCS), which aims to summarize a dialogue with several inserted stickers. The authors also build a new dataset for this task by first sampling informative sentences from a Chinese corpus, asking GPT-4 to complete the dialogues based on the initial sentences, and finally manually selecting and inserting stickers into the dialogues. The authors benchmark multiple summarization systems on this dataset and propose a TEI method that can help improve performance.

**Strengths:**

1. The major strength of this paper is that the construction process of the newly proposed dataset is elaborated. The authors take many important factors into account, like the privacy issue, how to select good initial sentences, etc. They also have well-performed human quality checks in every step of this pipeline. All of the above makes the new dataset very reliable, and it contributes significantly to the multi-modal dialogue community.
2. Many meaningful statistics for this dataset are provided so that we can have a direct understanding of how this new dataset differs from other related ones.
3. The paper is well-written and easy to follow.

**Limitations:**

1. One major weakness of this paper is that the authors do not include a sufficient number of models in their benchmark. For example, it is necessary to include VLM models like LLaVA, Qwen-VL, etc.
2. The proposed method, TEI, is quite trivial. I think it is more like an oracle one since it utilizes extra information generated by an advanced VLM (GPT-4-V), so the improvement is predictable.
3. Why, in Table 4, do the CLIPScore/RefCLIPScore not significantly drop when the stickers are switched to random ones, if we assume the original ones are highly related to the text?

**Suitability:**

3

---

### Official Review · Reviewer_wL7c · 2024-05-29

**Rating:** 5
**Confidence:** 3

**Summary:**

The paper proposes a new task called Multimodal Chat Dialogue Summarization Containing Stickers (MCDSCS) and introduces a corresponding dataset. The task focuses on generating text summaries for chat dialogues on social media that include both text and sticker images. The authors construct the MCDSCS dataset by generating dialogues using GPT-4, manually inserting sticker images into the dialogues, and generating summaries using GPT-4-Vision. The dataset consists of 5,527 Chinese multimodal dialogues containing 14,356 different sticker images. The authors also propose a novel method called TEI (Text descriptions of Emotions and Intentions) to incorporate the visual information from stickers into the summary generation process.

**Strengths:**

1. The paper addresses a new and important task in multimodal dialogue summarization, which has not been extensively explored in previous research. The MCDSCS dataset is the first of its kind, providing a valuable resource for this task.
2. The authors employ a clever approach to automatically generate dialogues and summaries using large language models, which can be scaled to construct larger datasets with minimal manual effort.
3. The proposed TEI method is intuitive and effective, as it addresses the key role of stickers in conveying emotions and intentions in social media conversations.
4. The paper provides a thorough evaluation of the proposed method against various baselines, including extractive algorithms, generative models, and even the powerful ChatGPT model.

**Limitations:**

1. The paper claims to have very good results. While they have done experiments with chatGPT, they did not use any of the SOTA and easily available baselines like CogVLM, LLaVA, etc.
2. The paper does not clearly define or justify the choice of evaluation metrics used, which can make it difficult to compare the results with other studies or tasks.

**Suitability:**

3

---

### Official Review · Reviewer_vRpX · 2024-06-07

**Rating:** 4
**Confidence:** 3

**Summary:**

This paper proposes a multimodal chat conversation summary with stickers (MCDSCS) task and dataset. The dataset consists of 5527 Chinese multimodal chat conversations and 14356 different emoticon images. The author intersperses emoticons in each text conversation to reflect the real social media chat scene, providing relevant data for multimodal conversations.

**Strengths:**

1.Presentation: The purpose of the article is well presented. Currently widely used dialogue systems, such as GPT-4, lack dialogue summary datasets that include both expression images and text. The necessity is well argued. The structure and layout of the paper are good, and the stickers are vivid.

2.Novelty:  By expanding stickers in existing conversations, the single-modal dialogue corpus is expanded to a multimodal dialogue corpus, and the dataset construction method is novel.

**Limitations:**

1.Novelty:  The TEI method proposed in the paper has no formal expression such as formulas, nor does it have a model architecture flowchart or a comparison with other existing methods. The proposed model solution may not be novel enough.

2.Experiments: There is little discussion on experimental analysis, and there is a lack of hyperparameter experimental analysis and related theoretical descriptions. For example, last year's Gemma model and this year's Llama-3, it is recommended that the author can demonstrate the superiority of the method in the latest model.

3.Flexibility:  Most of the image examples cited in the paper have text descriptions. For the dataset builders, if there is no text expression, there is a high probability that errors will occur when embedding the dialogue. If only an OCR is used for text recognition, will it be easier to recognize, and then retrieve the text from the image and text pairs with text descriptions, so as to achieve the situation where the image and text appear at the same time in the dialogue? The practicality of building from the dataset does not seem strong.
The dataset proposed in this paper adds 147,000 stickers to a total of 5.5k dialogues. Objectively speaking, the amount of data is insufficient for a large model dialogue corpus, especially for the training stage of a large multimodal dialogue model.

4.Reproducibility：The supplementary materials of the paper do not provide simple case codes, such as trained model files, and it is questionable whether they can be reproduced.

**Suitability:**

2

---

### Meta-Review · Area_Chair_VfGo · 2024-07-02

**Recommendation:** Accept (Poster)
**Confidence:** 5

**Metareview:**

This paper introduces a novel Multimodal Chat Dialogue Summarization Containing Stickers (MCDSCS) task and dataset, addressing an important gap in current dialogue summarization research. The reviewers generally praised the novelty and importance of the task, the thorough dataset construction process, and its potential impact on future research. While some limitations were noted by different reviewers, including the need for comparisons with more recent models and questions about the TEI method's novelty (reviewer vRpX, Vqn3), the authors' rebuttal appears to have addressed many concerns satisfactorily. Notably, Reviewer 9wao upgraded their assessment from Borderline Accept to Weak Accept after the rebuttal, recognizing the dataset's importance for future research. The primary strength of this paper is its dataset contribution. I believe that it has the potential to be helpful for the ACM MM community. Given the overall positive reception, the contribution of the dataset, and the authors' good rebuttal, I recommend accepting this paper for publication at ACM Multimedia.